# Review: Continuous Manufacturing of Small Molecule Solid Oral Dosage Forms

**DOI:** 10.3390/pharmaceutics13081311

**Published:** 2021-08-22

**Authors:** John Wahlich

**Affiliations:** Academy of Pharmaceutical Sciences, c/o Bionow, Greenheys Business Centre, Manchester Science Park, Pencroft Way, Manchester M15 6JJ, UK; john.wahlich@btinternet.com

**Keywords:** continuous manufacturing, drug products, control strategy, traceability, residence time distribution, real-time release testing, process analytical testing, process validation and verification, modelling

## Abstract

Continuous manufacturing (CM) is defined as a process in which the input material(s) are continuously fed into and transformed, and the processed output materials are continuously removed from the system. CM can be considered as matching the FDA’s so-called ‘Desired State’ of pharmaceutical manufacturing in the twenty-first century as discussed in their 2004 publication on ‘Innovation and Continuous Improvement in Pharmaceutical Manufacturing’. Yet, focused attention on CM did not really start until 2014, and the first product manufactured by CM was only approved in 2015. This review describes some of the benefits and challenges of introducing a CM process with a particular focus on small molecule solid oral dosage forms. The review is a useful introduction for individuals wishing to learn more about CM.

## 1. Introduction

This review on the Continuous Manufacture (CM) of small molecule solid oral dosage forms provides a basic introduction to the key elements of a CM process, gives the references to some of the key publications on the subject and mentions some of the recent developments and future opportunities. For a more in depth understanding of the subject, see Kleinebudde et al. [1].

CM has been defined to ‘be a process in which the input material(s) are continuously fed into and transformed within the process, and the processed output materials are continuously removed from the system’, where ‘system’ is defined as an integrated process that consists of two- or more-unit operations [2].

CM may be applied both to the drug substance manufacture and to the drug product. Developing a drug substance CM process is more complicated than a drug product one. The linkage between the two CM processes is described as end-to-end or E2E manufacturing (Section 5.2) (Figure 1).

The first regulatory approval of a drug product made by CM was in 2015. Since then, take-up by industry has been quite slow despite encouragement from the regulatory agencies. To date, there have been seven approvals (see Table 1). Some of these products use a combination of CM and batch manufacturing.

A research report put the market size for CM at 2.3B USD in 2018 with the potential to grow at a CAGR (Compound Annual Growth Rate) of 8.8%, reaching 3.8B USD in 2024 [6].

This paper will discuss the manufacture of tablets by direct compression (DC), dry granulation (roller compaction) and wet granulation (WG). These processes are listed in increasing levels of complexity. Some of the unit operations making up the processes are inherently continuous (i.e., tablet compression), while others are quite difficult to make continuous (e.g., tablet coating). CM may consist of a few unit operations linked together up to a fully integrated drug product system and on to an E2E system (of which few details have been published).

Regulatory authorities (EMA, MHRA, FDA and PDMA) have all been strong advocates of CM in particular for its benefits to improve the quality and control of drug products and to produce a more flexible supply chain. The industry has been slow to adopt the approach mainly due to its conservatism, the cost implications and the potential difficulty of registering CM products. Regulators have suggested that CM is suited to NCEs requiring new facilities or to established products with expanding markets. Most investment in CM has been made by Big Pharma working with some academic and not-for-profit centres. There is limited information on CDMO (Contract Development and Manufacturing Organisations) involvement in CM, although Patheon (Thermo Fisher Scientific), Catalent and Aesica have been mentioned as having CM capabilities [7]. The drivers for generic companies to be involved are different to those of pharma companies, and the initial investment and technical requirements for data handling, etc. make their involvement less likely. A model where there are hubs between academic centres and CDMOs to develop CM processes may be the way forward [8].

## 2. Background

The benefits of adopting CM are listed in Table 2.

Janssen gained the following benefits from converting Prezista^®^ from batch to CM. Seven rooms were used to manufacture the batch product and only two for the CM process; batches took 2 weeks to produce and CM only 1 day. In 2017, Janssen planned to manufacture 70% of their ‘highest volume products’ by CM in 8 years, increasing yield and reducing waste by 33% and reducing manufacturing and testing cycle time by 80% [12].

These benefits need to be balanced against the challenges of adopting CM, as given in Table 3.

It will not be possible to put together a business case to adopt CM for all drug products. For products with high demand, the argument might be based on the higher quality and efficiency and the elimination of the starts and stops in a batch process; for low-demand products, it might be the increase in the product shelf life, elimination of over production and inventory holds; for volatile-demand products, it might be the avoidance of shortages by rapid response to demand [15]. 

Convincing senior managers to make the initial investment is one of the biggest challenges. Although the pharma industry is not a major user of energy or the creation of pollution, an emphasis on the potential for a CM process to be net-zero could provide an extra ‘carrot’ in company governance. The need for game-changing incentives for the business as a whole to make the change to CM is also advocated. Examples of regulatory incentives are those used to encourage the development of paediatric products or to speed products to the market by designating them as ‘Breakthrough Therapies’. Tax incentives have created manufacturing hubs in certain countries (e.g., Singapore, Ireland and Puerto Rico).

Examples of the incentives that might be considered to encourage the take up of CM [5] are:

Tax incentives
For investment in pharmaceutical manufacturing innovation, including CM in R&D and production.For the manufacturing of products using CM processes.

Regulatory incentives
Expedited approvals (similar to break-through designations) for products using CM for new products, generics or those moving from batch processes.Some form of certification for companies where the increased rigour and control in CM process allows simpler and faster regulatory approvals.

A report issued by CMAC and PWC entitled ‘Business Case Insights for Continuous Manufacturing’ [16] provides a useful summary of the arguments for introducing CM, and a book chapter by authors from the University of Cambridge presents the business case from a supply network perspective [17]. 

A poll was taken at the recent UK-focussed Virtual International Symposium on the Continuous Manufacture of Pharmaceuticals (18 February 2021) [18] (see Section 8), and the top two activities that would accelerate the adoption of CM were given as: the availability of skills to develop, implement and work (CM technologies) followed by the ability to share experiences and collaborations to de-risk and demonstrate technology applications and regulatory pathways (e.g., mock submissions). In the same poll, the responders were split 47:53 (yes:no) as to whether they saw regulatory approval as a barrier to the adoption of advanced manufacturing technology. 

## 3. Regulatory Aspects of CM

### 3.1. International Conference on Harmonisation (ICH)

Quality by Design (QbD) is a systematic scientific and risk-based approach to pharmaceutical development. The following ICH documents provide high-level guidance to the scope and definition of QbD as it applies to the pharmaceutical industry.

Although not specifically mentioning CM in relevant ICH guidelines (Table 4), the enhanced product and process understanding obtained from CM and the ability to adopt advanced manufacturing controls to improve the quality of the drug product means that CM ‘is a true and complete representation of QbD’ [19].

A specific ICH guideline on CM has been discussed for some time, and the business plan for ICH Q13 was formally endorsed in November 2018 [24]. A draft of ICH Q13 ‘Continuous Manufacturing for Drug Substances and Drug Products’ is under discussion, with a Step 1 document expected to be ready by June 2021 [25].

### 3.2. Major Regulatory Agencies

Matsuda has summarised the global regulatory landscape as it relates to CM and includes a table comparing the specialised CM teams, their purpose, how to contact them and what activities they have underway to support CM [26].

#### 3.2.1. FDA

Ever since the issue of Pharmaceutical cGMPs for the twenty-first century in 2002, the FDA has been encouraging the industry to take a continuous improvement approach to pharmaceutical manufacturing [27,28]. The aim is to modernise the supply chain, enhance the robustness of the manufacturing process and thereby reduce product failures and enhance product quality. To create, in the words of the FDA, ‘An agile, flexible pharmaceutical manufacturing sector that reliably produces high quality drugs without extensive regulatory oversight’ [9].

CM meets the intentions of these objectives, and hence the FDA has always been incredibly supportive of this manufacturing approach. In presentations at the ISCMP meetings and elsewhere, Janet Woodcock (now Acting Commissioner of the FDA) has emphasised this support. The alignment with QbD and the fact that CM can be implemented within the existing regulatory framework is a consistent message from FDA. 

The FDA published a paper in 2015 co-authored by one of the leading lights in CM at the FDA, Sau ‘Larry’ Lee, which detailed the features and benefits of CM [3]. The vision in this paper extended to an end-to-end CM process. A further paper [29] described the emerging technology as a key enabler for modernizing pharmaceutical manufacturing and set out five FDA initiatives to achieve this. 

While appreciating the benefits of CM, the FDA also recognised the challenges and initiated the Emerging Technologies Team (ETT) in 2014, where companies planning to submit an application including emerging technology could have early discussions with a team of experts to provide support and address concerns [30]. As of 2019, the ETT program had conducted over 60 interactions with industry, of which a third were CM-related [7]. The FDA also issued a public docket to invite discussion of issues related to the adoption of CM in the pharmaceutical industry [31].

The draft guidance for industry ‘Quality Considerations for Continuous Manufacturing’ [2] states that FDA expects the adoption of CM will reduce drug product quality issues, lower manufacturing costs and improve the availability of quality medicines to patients. The guidance covers all the quality considerations of particular relevance to CM.

Other relevant FDA guidelines:Guidance for Industry PAT-A Framework for Innovative Pharmaceutical Development, Manufacturing, and Quality Assurance (including real-time release) [32],Guidance on Process Validation: General Principles and Practices [33].

#### 3.2.2. MHRA

Samantha Atkinson is MHRA’s Chief Quality and Access Officer and spoke at the 2021 Virtual International Symposium on the Continuous Manufacture of Pharmaceuticals [18], on ‘Innovation Adoption and Uptake—If Not Now When?’, indicating the MHRA view that this is an ideal time to be adopting new manufacturing approaches. She stated that MHRA is changing from a controlling regulator to an enabler, with innovative products and processes being one of the three pillars of this transformation. 

The MHRA is linked with regulatory agencies in Australia, Canada, Singapore and Switzerland via the ‘Access Consortium’, working together to promote greater regulatory collaboration and alignment of regulatory requirements. It is also part of the wider Pharmaceutical Inspection Co-operation Scheme (PIC/S) which is a non-binding arrangement between 54 regulatory authorities aimed at harmonising GMP inspection requirements. These interactions should help ensure a more consistent approach to the review of CM submissions.

The MHRA Innovation Office [34] is the point of contact, and innovation in manufacturing processes is one of its top interest areas. 

#### 3.2.3. EMA

Dolores Hernan of the EMA, speaking at the third FDA/PQRI Conference on ‘Advancing Product Quality’ in 2017 [35], reiterated that regulators are supportive of innovative manufacturing and, while the EMA offer no specific guidance, the current regulatory framework is adequate to allow CM. In her presentation, she said that two CM applications had been made to EMA.

The EMA has formed two teams that can provide support to the implementation of innovative technologies: the PAT team [36] and the Innovation Task Force [37]. The latter covers both emerging therapies and technologies and brings together experts in quality, safety, efficacy, pharmacovigilance, scientific advice, orphan drugs and good practices compliance, as well as legal and regulatory affairs. As with similar groups in other regulatory authorities, early dialogue is encouraged during CM development. 

Other relevant EMA guidelines:Guideline on Process Validation [38],Guideline on Real Time Release Testing [39],Guideline on NIR (Near Infra Red spectroscopy) [40].

#### 3.2.4. PMDA

The Japanese regulatory authority CM activities are led by Yoshihiro Matsuda, who summarised their activities in a presentation at the ISPE CM Workshop [41]. As with other authorities, the PMDA are highly supportive of CM. They established the Innovative Manufacturing Technology Working Group (IMT-WG) in 2016 to facilitate the introduction of innovative manufacturing technologies and approved a submission using CM for a DC tablet product in 2018. Their views on applying CM to the manufacture products (accepting that they have limited experience in CM and hence the document is a draft) were set out in a paper [42], which includes specific comments on control strategy, batch definition, validation and stability testing. They have also published two guidelines, ‘Points to Consider Regarding Continuous Manufacturing’ [43] and, together with industry partners in the AMED group (see Section 8.2.10), ‘State of Control in Continuous Pharmaceutical Manufacturing’ [44].

### 3.3. Other Regulatory Considerations

Despite comments from the regulatory authorities that existing guidelines support the introduction of CM, there are some aspects that require further detail, and hopefully, ICH Q13 will provide this clarity.

Current guidelines require testing of three batches representative of commercial-scale manufacture for process validation and stability testing. The current definition of a batch can be applied to CM. Unlike a batch process where batch size and equipment types will change from development to commercial scale, in CM, the same equipment may well be used at both stages, with the process simply being run for longer periods to achieve commercial volumes.

The FDA divides process validation into three stages: stage (1) design of the process and establishing a control strategy; stage (2) process qualification, where the process capability is evaluated to ensure reproducible commercial manufacture, including (i) design of the facility and qualification of the equipment and utilities and (ii) process performance qualification (PPQ) and stage (3) process verification, assuring a state of control during routine production [33]. These guidelines require that PPQ be conducted under cGMP.

In the case of CM, where the operations performed in development (not necessarily under cGMP) essentially mimic commercial production (usually differing only in the length of the process), this restriction is a limitation. This seems to have been appreciated in the FDA guide on CM [2], which does not explicitly require PPQ conducted to cGMP. However, there is still the requirement that ‘the design of the initial PPQ study to examine a run time or manufacturing period should be representative of the intended commercial run time for the initial product launch’. The PMDA also require that batch sizes should be established, taking into consideration the operability of the equipment over an extended period and that three batches are required for process validation [41]. 

These requirements seem excessive given the accumulation of data and process understanding implicit in the operation of a CM process and obtained throughout development using the same equipment and seem to counter the benefits of CM to achieve a lean technology transfer for fast market access. 

Continuous process verification is an alternative to traditional process validation in which a manufacturing performance is continuously monitored and evaluated [20], and PAT tools are key enablers in this [38]. The standard for a CM process will include process monitoring and control capabilities which de facto imply continuous process verification [4]. The use of CM from early development through to commercial manufacture with the elimination of traditional scale-up allows the generation of a data-rich environment with early implementation of process validation and process improvement. This provides the opportunity to introduce process verification as an alternative to traditional validation approaches [19,45].

Interestingly, with regard to stability testing requirements, both the FDA and PDMA accept samples taken from shorter CM runs, provided these represent the commercial manufacturing methods and processes. The requirement for three batches remains and may be achieved by three CM runs using three API batches or from a single run where manufacturing variability is captured (e.g., by introducing different batches of input material(s) in a sequential manner). It is important that samples for stability testing are collected that are representative of the process and, in the latter example, capture the introduced variability (i.e., are taken at appropriate times during the CM run) [2,42].

## 4. Quality Considerations of CM Processes

The quality requirements of a pharmaceutical process as embodied in ICH Q8 (R2), Q9 and Q10 apply equally to batch and CM processes. However, there are additional quality-related considerations specific to a CM process.

### 4.1. Control Strategy

A control strategy is designed to ensure that a product of the required quality will be produced consistently [20] and is a fundamental part of the QbD approach [46]. Maintenance of the quality of the product produced in a CM process requires the development of a robust control strategy. This should be designed to control the quality of the product in response to potential variations in the process, equipment conditions, incoming raw materials or environmental factors over time [3]. Control strategy implementations can be categorised into three levels:

Level 1 (also known as a performance-based approach) makes full use of the real-time monitoring and feedback that can be built into a CM process. Thus, Process Analytical Technology (PAT) tools and other sensors monitor the quality attributes of materials in real-time. Process parameters can then be adjusted in response to changes in the CQAs (Critical Quality Attributes) with the use of feedback (adjusting process parameters of manufacturing steps before the detecting sensor) or feed-forward (after the sensor) controls to adjust the process and ensure the product remains in specification. ‘Focus on the final critical quality attributes and let anything else change if it has to change’ might sum up this approach (Prof Zoltan Nagy of Purdue, quoted in [8]). This level of control requires a high degree of product and process knowledge, as the controls require an understanding of the linkage between the material attributes and process parameters. This is the hardest control strategy to develop but offers the greatest benefits.

If the understanding between these factors translates to knowledge of how they influence each aspect of the final product specification, then there is the potential to introduce Real Time Release Testing (RTRT) to remove the requirement for any end-product testing. There have been a limited number of approved products in which RTRT has been adopted. 

Level 2 control (a parameter-based approach) makes use of the design space concept and end-product testing. Adjustments to the critical process parameters based on changes to the critical material attributes can occur within a defined range based on an understanding of how these impact the critical quality attributes of the final product. This can be applied to both a batch and to a CM process. However, in a CM process, adjustments can be made upstream as changes in the material attributes or process parameters are detected.

Level 3 control relies on tightly constrained material attributes and process parameters, and there may be limited understanding of how these influence the quality of the final product. There is extensive end-product testing to ensure the final product specification is met. This level of control is more applicable to a batch process as it is unable to cope with any transient disturbances in a CM process. 

Although a Level 1 control strategy is desirable for a CM process, the control level can evolve through the drug product lifecycle, and hybrid approaches can also be developed [4].

A batch process operating according to a Level 3 control strategy can be said to be operating under a steady-state, whereas a CM process operating according to Level 2 or Level 1 control is operating under a state of control [22]. Matsuda of the Japanese PMDA has presented a diagram showing how different phases of a CM process relate to the control state [44].

Examples of how control strategies can be developed are given in detail in a paper from the AMED group (see Section 8.2.10). Note that in this paper, they reverse the numbering of the control strategy levels [47] and for a twin-screw wet granulation CM process in a paper by Pauli et al. [48].

### 4.2. Batch and Lot Definitions

The standard definition of a batch as set out in US 21 CFR 210.3 is ‘a specific quantity of a drug or other material that is intended to have uniform character and quality, within specified limits and is produced according to a single manufacturing order during the same cycle of manufacture’. This definition applies as readily to a CM process as to a batch process. ICH Q7 states, ‘In the case of continuous or semi-continuous production, a batch may correspond to a defined fraction of the production. The batch size can be defined either by a fixed quantity or by the amount produced in a fixed time interval’ [49]. Batch refers to the quantity of material and does not specify the mode of manufacture [9]. A lot is a specifically identified portion of a batch having uniform character and quality within specified limits.

A batch in CM can be specified as follows [4]:The amount of material produced within a specified period of time (the most common definition);A fixed quantity of raw material processed.Or by operational considerations such as:The input API batch size;The length of an operator’s shift.

One of the main benefits of CM is the relative ease (compared to a batch process) in which the batch size can be modified [47] by:


Increasing the run time (within validated limits). Consideration needs to be given to issues such as the build-up of material during longer runs (in transfer lines, filters, granulator barrel walls, punch surfaces, etc.), equipment overheating, etc.Parallel operation of multiple items of kit. Need to assure the uniformity of output from each process.Increasing the throughput of the process. This has an impact on the process dynamics and may affect aspects such as sampling frequency and size for PAT equipment, Residence Time Distribution (RTD) and material diversion (see Section 4.3) and will likely require that the control strategy is rebuilt.Change of size of equipment. Also, likely to require a completely different control strategy and complete re-validation


Changing the run time is by far the most common approach and allows the same equipment to be run for a short time for development purposes and for a longer (validated) period for commercial scale.

All regulatory authorities require that the definition of a batch be specified before a production run commences (see for example [2]).

### 4.3. Traceability and Residence Time Distribution (RTD)

For a continuous manufacturing process, understanding the process dynamics of how a material flows through the process is important with respect to material traceability (the ability to preserve and access the identity and attribute of a material throughout the system) [19]. Likewise, tracking raw and intermediate materials to the final product is essential for understanding how material and process variation propagates through the process. For example, knowledge of the portion of a continuous run where a specific raw material lot was used might lead to the assignment of a new lot number to the final product batch [50]. If a disturbance in the CM run affects the quality of an intermediate stage or final product, then knowledge of how it propagates will be essential to know when and for how long to divert affected material.

Unlike a batch process where the location of the affected material is constrained, in a CM process, it will be located over a stretch of the process determined by the process dynamics. The RTD is defined as the probability distribution of time that solid or fluid materials stay inside one- or more-unit operations in a continuous flow system and is a fundamental chemical engineering concept [51]. 

Each unit operation and specific process has its own unique RTD. RTDs may be combined through convolution to show the effect of two- or more-unit operations. A steady-state RTD is one where a process is operating as normal, and a dynamic RTD is one where the process has been disturbed (during start-up or shut-down, for example). Each unit operation needs to be modelled to determine its RTD. Two RTD models that can be applied to a drug product CM process are one where the unit operation is considered to operate under plug flow reactor conditions (PFR model) and a second where it is modelled as if it was a continuous stirred tank reactor (CSTR). These two models reflect the two extreme RTD profiles in a unit operation. (A third type of model, the laminar flow reactor (LFR) approach, is more relevant to liquid-based CM systems.) Combinations of the two models can be used, and other models (such as the axial dispersion model) are available [51]. The model parameters can be determined by fitting them to the actual concentration of API or of a tracer as determined by continuous online measurements using NIR, UV or visible spectroscopy (if a coloured tracer is used). The material to be traced can either be pulsed into the process or changed in a stepwise manner and the output monitored. A multi-pulse approach can provide less variable data but requires longer to determine and uses more material. The RTD model which best describes the process is then selected [52].

The RTD model can be affected by the formulation, equipment and processing conditions (including throughput) and needs to be checked if any of these change (including any controls applied by the CM process). If a tracer is used, its properties must closely match those of the API or material it is tracing, and it must not affect the flow properties. An example of RTD applied to a DC CM process gives more details [53]. Two papers aim to focus thinking on the development of standard methods for conducting, interpreting and using RTD results in CM processes [54,55].

### 4.4. Process Disturbances and Diversion of Material

A CM process can be subject to disturbances, and if these cannot be accommodated within the process adjustments allowed by the control strategy, then the material must be diverted to waste or re-work, or in a worst-case scenario, the process must be shut down. A CM process for a direct compression product is likely to be classified as highly capable (Process Capability Index Cpk > 1.33), and as such, the biggest risks are drift or special causes (unexpected disturbances). Control strategies need to be developed to mitigate these [56].

Two obvious disturbances are during start-up and shut-down when the process has not yet achieved or is leaving the state of control. Other disturbances may occur: during hopper filling; if PAT or processing equipment malfunctions; due to unexpected changes to material attributes or build-up of material in the system, etc. These disturbances can cause a deviation from a material’s quality attributes. There are multiple approaches to detect these deviations and apply controls:Deviations that fall outside pre-set ranges of material attributes and process parameters as determined by development studies (e.g., response surfaces, design spaces);Prediction of quality attributes from process modelling with real-time inputs of process parameters and measurements;Direct measurement of quality attributes in real-time (use of PAT).

Any of these three approaches may be used individually or in combination. Response to these disturbances might be feed-forward or feedback control. However, if this does not bring the quality attributes back under control, then the material may have to be diverted [56]. 

The extent of material to be diverted (so-called ‘fencing’) depends on the duration, frequency and severity of the disturbance and the mixing patterns of the system [3]. RTD has been used to more accurately quantify the amount of material to be diverted during a disturbance and hence reduce waste [57]. It is preferable to divert material as close to the point of disturbance as possible. For a direct compression process, diversion can occur either after the point of mixing or after compression, and the advantages and disadvantages of each approach have been described [56].

FDA guidance states that diversion of material during start-up and shut-down and because of temporary process disturbances is an acceptable practice providing it conforms to established procedures. The process shutdown procedure should include what needs to happen for the shut-down of each of the unit operations and whether the shut-down is permanent (e.g., a clean down of the equipment is required) or temporary. In the case of the latter, factors such as the stability of materials held up in the process, any specific cleaning requirements, etc. need to be defined before the process can be restarted. Diversions caused by unexplained disturbances need to be investigated and may affect the release of the batch [2]. 

When the CM process is operated at a small scale (short run time), for example, if CM is being used during development or for a niche product, the diversion of material during start-up can be significant and expensive. A variable control strategy has been suggested to minimise waste [58]. This has been applied to a semi-continuous segmented fluid bed dryer to adjust the drying time during the initial warm-up of the equipment. This is built into the process during development as standard PAT feedback controls were not fast enough to make the necessary changes.

### 4.5. Real Time Release Testing (RTRT)

RTRT is a system of release that gives assurance that the product is of intended quality, based on the information collected during the manufacturing process, through product knowledge and based on process understanding and control [37]. Originally used for the parametric release of sterile products, the monitoring of a continuous manufacturing process using PAT tools can generate a large amount of real-time process and quality data during production, which can be used to support RTRT. Although RTRT is not a regulatory requirement for the implementation of continuous manufacturing processes, it is encouraged and could be applied to some or all the finished product quality attributes tested for release of the batch [2]. Examples of quality attributes and considerations for RTRT implementation are given in this FDA-authored paper for identity tests, tablet assays and content uniformity and for models serving as surrogates for drug release testing. 

A recent review paper on RTRT describes the critical quality attributes which might be monitored at different stages with examples in CM processes to produce tablets by direct compression, wet granulation and dry granulation (roller compaction) [59]. Methods used to make quality-based control decisions during RTRT are referred to as primary methods and are considered as high-(impact) risk methods according to the risk impact matrix as described in ‘ICH Quality Implementation Working Group Points to Consider (R2)’ [23].

The review paper by Markl et al. offers ideas as to how each CQA might be monitored either by direct measurement or using a multivariate model to relate the measurement to an attribute. The authors stress that the relevance of a measurement needs to be assessed on a product-by-product basis. CQAs for a solid oral dosage form are typically those relating to purity, strength, drug release and stability. Examples of the measurement techniques used for each attribute follow [59].

#### 4.5.1. Blend Mixing and Tablet Content Uniformity

Typically uses NIR and/or Raman spectroscopy. Measurement typically occurs on the blend prior to tabletting. Bhaskar et al. [57] note there is no sensor currently available that can accurately measure tablet potency and/or drug concentration of tablets in real-time, although Markl et al. [59] suggest a sensor incorporated in the feed frame of the tabletting machine.

The FDA has received requests to use a parametric approach to justify acceptable blend or tablet content uniformity. The proposal is that API feeder mass flow rate coupled with historical data on acceptable quality can be used to replace PAT testing of API concentration. The FDA does not accept this approach as an alternative to a validated model to quantitatively link material properties with upstream and downstream process conditions to the final product quality attributes [25].

#### 4.5.2. Moisture Content 

This is of relevance in a wet granulation process, and NIR or microwave resonance technology are typically used.

#### 4.5.3. Solid State Chemistry 

This is most likely a CQA following a wet granulation process. NIR and/or Raman can be used.

#### 4.5.4. Drug Release (Dissolution)

This is the hardest CQA to predict. Measurements of granule size and density (for a granulated product), tablet strength, weight and thickness, porosity and coating thickness can potentially be combined to predict drug release. This will be easier for a BCS Class 1 product where disintegration is the CQA. Markl et al. [59] note that there is a need for better methods to determine tablet porosity. 

Pawar et al. [60] were the first to publish a dissolution prediction method. They used NIR measurements and a Weibull model to predict the dissolution of acetaminophen tablets from a DC CM process. A combination of NIR and Raman measurements and an artificial neural network has been used to accurately predict the dissolution of an extended-release tablet formulation [61].

Bawuah et al. [62], on the other hand, used terahertz spectroscopy to determine tablet porosity (without the need for a model) and found a good correlation with tablet dissolution (time to 50% release), disintegration time and tablet strength and suggested it might be applicable to immediate-release products where disintegration is used to monitor drug release. 

The Orkambi^®^ CM product is a fixed-dose combination of lumacaftor and ivacaftor 200 mg/125 mg. It was approved by the EMA in 2015 and used RTRT, including for confirming the dissolution of both drugs [63]. Details of what technologies were used were not given.

Prezista^®^ CM product was originally approved by FDA in 2016 [64]. In 2017, C-SOPS (see Section 8.2.3), working with Janssen, filed a post-approval change covering RTRT for identity, assay and dissolution, which was approved by the FDA, making it the first US product that could be released without end-product testing [65].

#### 4.5.5. Related Impurities

Synthetic related impurities can be monitored in the drug substance specification. If degradation subsequently occurs as part of the drug product manufacturing process, then current PAT methodologies are unlikely to be sensitive enough to detect and quantify these and hence allow RTRT release. If the API can be shown to be stable during the drug product manufacture, then related impurities need not be included in the release specification. Whether the regulators would accept RTRT if there is a small degree of degradation, within specification and proven to be consistent across several batches without requiring a PAT test, is unclear. 

### 4.6. Process Validation and Verification

The various guidelines issued by different regulatory authorities related to process validation and verification (see Section 3) and ICH Q8 (R2), Q9 and Q10 [18,19,20] are applicable to CM processes. For CM processes, the ability to evaluate real-time data to maintain operations within established criteria, to produce drug products with a high degree of assurance of meeting all the attributes they are intended to possess, is an integral element of process validation [2]. In CM processes, the three stages of process validation (Stage 1 Process Design, Stage 2 Process Qualification and Stage 3 Continued Process Verification (CPV) may be more interrelated and may run concurrently as the equipment used during development may be the same as that for commercial scale. Thus, several of the items in Stage 2 may be more appropriately performed in Stage 1. The process performance qualification (PPQ) aspects of validation in Stage 2 should reflect the length of the run at a commercial scale to detect process drift, equipment fatigue and material build-up. PPQ should include interventions that might normally occur during a run, such as feeder refills, PAT probe replacement, etc. 

Specific items related to CM which need to be considered are the residence times within the process and whether these are sufficient to achieve the desired outcome. For example, ‘is a homogeneous mix achieved in a blender or twin-screw granulator?’ and / or ‘is sufficient time allowed for a compressed tablet to relax before coating?’ The ability of the process to reach a state of control at start-up and switch from this at shut-down needs to be validated. 

A CM process is well placed to satisfy the requirements of Stage 3 validation as data should be collected continuously as part of the control strategy. This potentially provides an opportunity for CPV to replace some of the traditional aspects of process validation (see Section 3). 

### 4.7. Models Used in CM

PAT sensors used in a CM process either monitor process parameters or material attributes directly or use multivariate models to convert their data into relevant process information. Soft sensor models are predictive models where the value of a quality variable is not directly measured but is inferred from process data (e.g., models to predict dissolution performance) [66]. A CM process operating under a Level 1 control strategy likely makes use of several models. The models may be quite complex and may have inputs in addition to sensor measurements such as variables originating from raw materials, equipment parameters and the environment, to name a few. Validation of the models is included as part of process validation, and they require maintenance and are subject to the CPV requirements. 

Models can be low-, medium- or high-impact, depending on their use. Models which are directly related to the final product CQAs are high impact [23]. Most models used in CM as part of a control strategy will be high impact. Changes to a high-impact model typically require a regulatory submission. This can be a major issue as there are no common standards among the regulators on the validation, maintenance and update of models. Industry preference is that model maintenance is handled as part of a company’s Product Quality System (PQS) process [59]. There is a chance to harmonise these requirements [25] as part of ICH Q13 or to utilise the Post Approval Change Management Plan (PACMP) described in ICH Q12 [67], which allows for specific changes to be pre-described to regulators and agreement reached on the scientific approach and data expectations that will support the change. 

The complexity of the models used in a CM process and the requirement for their maintenance can make the transfer of these processes difficult if the receiving site does not have the requisite technical support. 

### 4.8. Data Requirements

PAT produces large amounts of data. There are quality and regulatory considerations relating to the need to store these data, including the use of Cloud storage and the application of AI (Artificial Intelligence) to process the data. Further discussions are needed between industry and regulators to agree on standards and regulatory requirements [25].

### 4.9. Summary

Figure 2 summarises the quality considerations of a CM process and how they evolve during development and into commercialisation. 

## 5. CM Equipment

### 5.1. Unit Operations

Several of the unit operations for tablet manufacture are intrinsically continuous such as mills, roller compaction equipment for dry granulation and tablet presses. Integrating these together and developing equipment for the non-continuous stages of a process is a relatively new development.

Continuous direct compression tabletting is one of the simpler processes and consists of dosing stations with multiple loss-in-weight feeders, a continuous blender to mix API with excipients, a second blender to add lubricants and a rotary tablet press. A dry granulation process uses a roller compactor and mill in addition to feeders, blenders and a tablet press. The additional equipment used in a wet granulation process (which is the most complex of the three tablet manufacturing approaches) includes, most commonly, a twin-screw granulator, usually with a mill and a dryer. Tablet coating, to date, has typically followed a batch process.

Readers are referred to Burcham et al. [10] and to Teżyk et al. [69] for a more detailed description of CM equipment for tablet manufacture and associated PAT.

#### 5.1.1. Feeders

The most common feeders are loss-in-weight feeders, where a hopper sits on top of a weighing platform, and powder is dispensed from the hopper by means of a screw mechanism, vibration, belt or rotary valve. Changes in the weight of the system are monitored by a controller, and the speed of the screw (as the most common feed mechanism) is altered accordingly. Alternatively, the feeder can operate in a volumetric mode where the screw speed is kept constant.

Fluctuations in feed rate are the most common causes of transient disturbances in a CM system. These may include feed fluctuations of non-cohesive powders due to the discontinuous nature of solids and screw patterns, downspout accumulation due to triboelectricity and feeder ‘bearding’ and fluctuations caused when the feeder needs to be refilled. Small disturbances can be compensated for by the feeder controls and necessary action taken downstream by the overall process control system. Larger pulses (such as might occur if a ‘beard’ of material breaks off from the feeder) have to be included in the risk assessment of the process and their impact assessed by RTD models, including the potential to divert affected material. During refill, the weight control from the feeder is lost, and it switches to volumetric mode. Advice is that refill should occur within 10 s and not when the feeder is less than 20% full [70].

The RTD in a screw feeder can be represented by a combination of plug-flow and mixed-flow models and can be affected by flow rate, hopper fill level and conditioned bulk density [71].

The use of microwave sensors to monitor powder flow during periods when the loss-in-weight feeders are being refilled has been proposed, and initial results were promising [72].

#### 5.1.2. Powder Blenders/Mixers

The three fundamental blending mechanisms are convection (the gross movement of particles either by the action of a paddle or by gentle tumbling), diffusion (intermingling of particles at the small scale, tends to be slowest and determines the blending time frame) and shear (involving the thorough incorporation of material). In a continuous blender, convection occurs in parallel with the primary flow of material through the blender; diffusion occurs perpendicular to this movement. Shear forces can be added to break up agglomerates and disperse particles.

There are two broad types of continuous blender: drum systems (outer housing rotated) and screw or paddle systems (when an inner shaft rotates). Different designs of these can enhance the blending (e.g., the addition of baffles). API and raw materials must be fed into the blenders at a controlled rate, and this is critical to the efficiency of the mixing [73].

Continuous mixers have been shown to produce a more homogeneous output compared with mixers operating in a batch mode [74] and also have the added advantage of the potential to add PAT. Ervasti et al. [75] considered whether one of the reasons for the slow take-up of CM might be due to concerns over which products it was most suited. On this basis, they studied the application of a CM direct compression process to low-dose API products, where achieving good homogeneity is a real challenge. They considered whether it might be possible to have a generic DC matrix (comprising lactose and microcrystalline cellulose with magnesium stearate as a lubricant), as the physical properties of the API added at low doses might not affect the overall properties of the mix, and studied this using two APIs (added at 3% loading) with poor flow and aggregation tendencies. They concluded that this approach seemed to work but would need studying for longer run times.

#### 5.1.3. Granulators

Twin-screw granulators (TSGs) (also known as twin-screw extruders) are used in CM wet granulation processes and are essentially small volume high shear continuous mixers. For dry granulations, roller compactors, which inherently operate in a continuous mode, are used and produce a ribbon of compacted material which is then milled. 

TSGs consist of a barrel in which two rotating screw elements intermesh (either co- or counter-rotating), mix and convey materials along the barrel. API and excipients are introduced via loss-in-weight feeders, and a granulating fluid is added. As the granulation progresses along the barrel, the powder blend undergoes wetting and nucleation, consolidation and coalescence, and attrition processes turn it into granules. TSGs are not typically used in batch processes unless it is to produce amorphous solid dispersions via melt granulation. In this use, the granulating solvent is replaced by a polymeric binder producing a highly viscose medium at elevated temperatures.

Advantages of TSGs in a CM process compared to batch granulation are a more homogeneous distribution of drug, excipient and binder solution, a reduced level of binder solution to achieve the desired granules and shorter granulation times. The latter benefit can be used to advantage, and granulation can be operated at higher temperatures should the API be thermally labile [76].

Glatt offers a single-shaft continuous wet granulator, which offers mid-shear mixing. Lodige’s CoriMix^®^ is a continuous ring layer versatile mixer which can also be used as a granulator.

A recent paper by Nandi et al. [77] discusses advances in TSG and includes details of applicable PAT, while a paper by Portier et al. [78] discusses TSG from the perspective of the formulation and raw material properties.

#### 5.1.4. Dryers

Dryers for continuous operation have been relatively late on the scene. They are typically sold as combination units coupled to TSGs.

A modified semi-continuous version of a fluid bed dryer (FBD), equipment typically used in a batch process, is the segmented FBD offered by GEA in its ConsiGma™ range. This has several chambers that rotate around a central air flow unit. These are sequentially filled from a continuous feeder (typically coupled to a TSG), and the material is fluidised. When drying is complete, the material is discharged, and the empty segment is refilled to achieve a continuous operation.

Glatt offer a fluid bed dryer that has four static chambers and a rotor blade that moves materials between the chambers to achieve a continuous operation. This can be fitted with a granulation process insert to produce a continuous low shear granulator. Their continuous fluid bed dryer (GPCG 2) has 10 chambers and can be coupled to a Thermo Fisher Pharma 16 twin-screw granulator. This set-up has been studied in detail and has been shown to achieve a very narrow RTD [58,79].

Lődige couple an LCF5 fluid bed dryer with their continuous Granucon^®^ granulation line. A screw device forces materials with up to 20% moisture content through the fluid bed dryer enabling dryer retention times to be reduced in continuous operation.

The system manufactured by Leistritz couples the TSG with a heated barrel and then to a flash dryer where the remaining liquid can be removed after discharge [76].

LB Bohle offer the QbCon^®^ continuous granulator dryer in which granules are placed on a vibrating conveyor and are subjected to a flow of hot air which gently fluidises them on a vibrating horizontal conveyor. They dry in a relatively short time (on average 60 s), which lowers the value of the RTD and thereby reduces the amount of any material that might need to be discarded and facilitates traceability [80].

Freund’s Granuformer^®^ combines a twin-screw extruder with a spiral dryer to give granules that offer properties producing stronger tablets that dissolve more rapidly than those produced by a high-shear mixer or by a fluid bed process [81].

#### 5.1.5. Coaters

Coating has proven to be the hardest tabletting unit operation to convert to CM. Tablets tend to relax after compression, and if coating occurs too soon, they can expand and crack the coat. Hence, a holding time is required before they can be coated satisfactorily [82]. This hold time is incompatible with the continuous nature of the CM process. In addition, coating quality (of particular importance if the coat is a functional one) is directly affected by the length of time each tablet is exposed to the coating process and hence a very narrow RTD window, ideally a spike, is required. This is very difficult to achieve in a continuous coating process [83]. Unsurprisingly, therefore, while a number of semi-continuous and continuous coating systems have been developed, there are few peer-reviewed publications on their use and limited recent developments.

LB Bohle was the first manufacturer to offer a semi-continuous coating machine (the Koco^®^) in 2011.

GEA’s ConsiGma™ continuous coater consists of two coating ‘wheels’ which work in tandem to coat semi-continuously; as one unit coats, the other is filling, and as one discharges its 3 kg load, the other begins to coat. Coating times average 5–10 min.

Continuous coaters from O’Hara Technologies Fastcoat™ or Thomas Engineering’s Flex CTC^®^ or Driam’s DriaConti-T pharma^®^ and IMA’s Croma^®^ coating system, introduced in 2018, operate in a similar fashion with lengthened coating pans divided into zones or chambers with one or more spray heads per section. The pans can be up to 15 feet in length with up to 7 chambers and up to 24 spray heads, each of which can be independently controlled. Control of the spray heads can minimise the waste during the start-up and shut-down of the coater. Tablets are introduced at one end using a gravity feeder and move along the pan. The coating is typically completed in <15 min, which compares with, on average, 40 min in a batch process. The manufacturers all claim narrow RTDs and the ability to fit various PAT devices to their systems. IMA’s Croma^®^ coater is designed to be coupled to a Prexima^®^ tablet press and can incorporate a buffer system to give tablets sufficient time to relax [84].

Colorcon market the Opadry^®^ QX non-functional coating material specifically designed for continuous coating systems. The material has a very low viscosity, which enables dispersions of up to 35% solids and results in faster application rates [85].

### 5.2. Integrated Continuous and End-to-End (E2E) Manufacturing

A company introducing CM might want to link various units together and, as their understanding of the process increases, extend these to achieve an integrated drug product manufacturing system or a truly E2E system with continuous API synthesis included [86].

Interestingly, Vertex’s approval of their Orkambi^®^ fixed-dose combination film-coated tablets prepared by wet granulation was for three different manufacturing sites, each operating slightly different combinations of batch and CM processes and end-product and RTRT release. One of the sites employed a twin-screw granulator fed by a batch blend, followed by stand-alone batch fluid bed drying. The second site had a continuous tabletting line that operated in a continuous mode from granulation to compression, with initial blending and film coating performed in batch mode. The third site used a system that operated in continuous mode from individual components feeding to film-coated tablets and was enabled with real-time release testing (RTRT) capability [63].

In addition to the various stand-alone CM units, which can be combined as required into an end-to end CM process, a number of suppliers offer packages.

The GEA ConsiGma™ range combines continuous wet granulation, segmented fluid bed drying and tablet compression into one production system with options to include additional feeders and blender units, mills, melt granulators and PAT tools. GEA probably have the most experience in CM. A ConsiGma™ wet granulation manufacturing line has been modelled, and global sensitivity analysis models are used to determine Critical Process Parameters (CPPs) that affect the process responses of interest. The liquid feed rate into the granulator, air temperature and drying time in the FBD were identified as CPPs affecting the tablet properties [87]. Merck Sharp and Dohme, together with GEA, ran a ConsiGma™ for 120 h, and during that time, produced 15 million tablets with a yield of 99.5% [88]. Pictures and diagrams of GEA’s ConsiGma™ equipment can be found in [77,78].

Glatt’s MODCOS^®^ modular continuous processing system couples their proprietary powder dosing, mixing, granulating and drying systems together with tablet presses from Fette and the Driaconti-T^®^ continuous coaters from Driam to produce a continuous integrated system. The modular nature of the system allows up to 10 feeders and 3 mixers to be combined in one process. A picture and diagram of this equipment are shown in [77].

A picture of LB Bohle’s QbCon^®^ equipment integrated into a linked CM system can be seen in [78].

The Bosch Xelum^®^ R&D system combines dosing (up to four feeders) and fluid bed granulation in one unit, which eliminates the need to transfer wet granules, which could be a source of disruption in a TSG continuous system. Bosch and RCPE announced a collaboration on CM in 2016 [89].

One way of partly linking the drug substance synthetic process to the drug product takes the output from the synthesis before isolation of the API and feeds this into the drug product manufacture. As an example, a 50% aqueous suspension of ibuprofen was fed into a TSG to produce a wet granulation which was then processed by CM [90].

In a project funded by the US Defense Advanced Research Projects Agency (DARPA), MIT developed a Pharmacy on Demand (PoD) unit 72 × 53 × 134 cm^3^ in dimensions which could produce tablets by direct compression from input APIs. They demonstrated the capability of the system with ibuprofen and diazepam [91]. Subsequently, DARPA announced an agreement with Continuity Pharma (a spin-out from Purdue University) to develop CM technology for the synthesis of APIs [92].

Novartis/MIT demonstrated the first truly E2E CM process from API synthetic intermediates through to the finished dosage form by producing aliskiren hemifumarate tablets in an operation that took 2 days to complete compared to at least 12 days for the batch process (not including off-line holding and transport times) [93]. Continuus Pharmaceuticals was established in 2012 as a spin-out from the Novartis/MIT collaboration and plans true E2E manufacture. They are working in collaboration with IMA, an Italian pharmaceutical equipment maker. In January 2021, they won a 69M USD contract from the US Government to onshore three vital small-molecule medicines using a GMP-certified E2E manufacturing facility [94].

The advantages of an E2E supply chain have been described in a paper resulting from the 2014 ISCMP Conference. The authors advise that it will be very technically complex [14].

### 5.3. Alternative Continuous Manufacturing Approaches

The equipment discussed above shows how the traditional process to produce tablets can be made continuous, however, alternative approaches are possible. Additive manufacturing (also referred to as 3D printing) has received considerable attention and has the added advantage over CM of offering dose flexibility and tablet customisability. A detailed discussion is beyond the scope of this review, and readers are referred to a recent review paper by Abaci et al. [95].

GSK has developed a semi-continuous manufacturing system called Liquid Dispensing Technology, which sprays API onto a placebo tablet core in a process similar to ink jet printing. Placebo cores (which might themselves be prepared in a CM process) with a slightly indented surface are fed on a conveyor under a dosing head, which squirts a solution or suspension of the API (in water or a solvent) onto the tablet surface. The tablets are then passed under a dryer, and the API amount is confirmed by a PAT sensor. The tablets may then be optionally coated. The technology is ideal for low-dose products and avoids the usual issues of poor content uniformity. As the API is handled in a liquid, it also minimises exposure to highly potent compounds [96].

## 6. Process Analytical Technology (PAT)

PAT sensors are positioned at key points in a CM process to determine the CQAs of intermediates and the finished product, including for RTRT. Data from the sensors are used as part of the control strategy for the process and can be used to alter the processing parameters to ensure the CQAs remain within acceptable limits [20]. PAT data can provide:Direct measurement of the CQA;Prediction of the CQA based on a first-principles model that is fed measurements of related variables and is running in parallel with CM unit operations;Prediction of the CQA based on an empirical or semi-empirical model (e.g., response surface map, chemometrics model) that is fed measurements of other variables;Operation of the CPPs to lie within a design space (ie a specified set shown in offline studies to provide assurance) [97].

The FDA has produced a guideline on PAT [32], and this describes the potential positioning of sensors:At-line: measurement where the sample is removed, isolated from and analysed near the process stream.On-line: measurement where the sample is diverted from the manufacturing process and may be returned to the process stream.In-line: measurement where the sample is not removed from the process stream and can be invasive or non-invasive.

Factors such as where to place the sensor in the process, probe depth and sampling area, sample size (representative of a unit dose) and sampling frequency all need to be considered together with data handling, actions to be taken if the sensor fails, calibration and maintenance (including probe cleaning particularly if positioned in-line). The sampling frequency is critical in CM. It needs to be fast enough to detect a transient change, to detect a process drift (collecting enough measurements for trend analysis) and to allow adequate assessment of the quality of a batch based on sound statistical criteria [19].

For guidelines on the application of PAT, see [98,99]. A recent paper lists the CQAs from each unit operation, which PAT techniques are best used to monitor these and has more details on each of the techniques mentioned here [4]. Another recent paper [68] discusses unit operations from a CPP perspective and describes the PAT tools that might be used. A paper by Markl et al. presents a detailed literature review of RTRT application in tablet manufacturing with details of the PAT technology used, point of testing and details of the analysis [59].

Raman and NIR are the most common spectroscopic sensors and can be applied to determine CQAs in almost all unit operations in CM. NIR, for example, has been used to determine API concentration, moisture content and prediction of granule size distribution, granule porosity, tablet disintegration time and friability. The techniques are fast and non-destructive. Each responds to different molecular characteristics so has different uses. The EMA has produced a guideline on the use of NIR [40], and the FDA has a draft guideline on the development and submission of NIR procedures [100].

Recent advances in the use of NIR and Raman include their determination of blend uniformity and content uniformity for low-dose formulations (Raman was more sensitive in one example, being able to monitor a 1% drug concentration), the determination of different CQAs from the same spectra (e.g., granule size distribution, moisture content and API concentration) and prediction of dissolution behaviour (mentioned elsewhere in this paper) [4].

The disadvantages of the use of NIR or Raman as PAT technologies are that the methods can be a challenge to develop and implement, and they require a high degree of overhead and technical resources to maintain the calibration models. For example, they may be impacted by changes in the particle size distribution of an excipient and often require regulatory approval when changes are made. Medendorp et al. [13] discuss these challenges, why they create a barrier and some of the potential ways of reducing the maintenance burden (covered elsewhere in this paper). They suggest the use of non-spectroscopic alternatives such as process signatures, temperature and weight measurements as alternatives to provide assurance of a state of control with much greater operational simplicity. Their example uses gravimetric measurements to determine API concentration and is applied to a continuous twin-screw wet granulation process. The basic premise is that the amount of API per tablet should be directly calculatable from the amount added into the process (monitored by the loss in weight feeder). In addition to ensuring that the feeder load cells are properly maintained, it is important to consider the precision of the material dispensing, the assurance of uniformity at both the granule and final blend stage, the evaluation of moisture content and the cumulative effect of all these items on the calculated blend potency. A very recent comment from the FDA suggests they might require some convincing before accepting this parameter-based approach [25].

Other authors have highlighted the use of UV-VIS as a monitoring technique for studying the output from twin-screw extruders. This is a much simpler and more sensitive technique than NIR or Raman. It can be used to study the RTD of a system with the addition of a marker and has the advantage that its sensitivity means that the marker (in the example, quinine dihydrochloride) can be added at a level that will not affect the process rheology (<100 ppm) [101]. Alternatively, UV-VIS can be used to monitor the drug (in this example, piroxicam) in the formulation directly [102]. In both these studies, the ColVisTec probe was used.

Other less often reported PAT techniques include the use of X-ray absorbances to monitor powder flow; ultrasound to detect tablet defects, strength and porosity; microwave technology to determine API concentration, density and moisture content of roller-compacted ribbons); terahertz spectroscopy for solid-state characterisation, coating thickness, porosity and dissolution prediction; infrared thermography to monitor roller-compactor ribbons (see Vanhoorne et al. [4] for details and further examples). Markl et al. [59] have commented on the need for better PAT technologies to determine tablet porosity.

The Research Centre for Pharmaceutical Engineering (RCPE), based in Graz, Austria, has been studying optical coherence tomography [70] and spun out a company Phyllon^®^ to market this. They were awarded a grant in 2020 by the FDA to explore its use for the real-time monitoring and control of the drug tablet coating process [103].

## 7. Material Property Requirements for Continuous Manufacturing

A CM process requires additional testing of the input materials compared to pharmacopoeial requirements due to the automated nature of their introduction into the process via feeder systems. Physical property testing usually performed on the API must be extended to the excipients. Attributes such as particle size distribution, shape, surface energy, wall friction, density, cohesivity, tribocharging potential, etc., which might affect their flow behaviour and segregation potential, need to be determined and specifications set. Inter-batch variability needs to be determined as this may affect the process when feed hoppers are refilled. Modelling methods can be used to analyse data from multiple materials and identify the key relationship between the materials, process and product [104].

Variations in material attributes can also affect spectroscopic methods and process models (RTD, etc.) [105]. Ideally, materials should be quantifiable by NIR or Raman.

An Academy of Pharmaceutical Sciences (APS) working group developed the Material Classification System (MCS) to identify the material properties and how they affect their suitability for different unit operations [106]. A second paper [82] includes a section on the application of the MCS to continuous processing. A table in this paper shows that while the properties of different batches of API are all suitable for producing tablets by DC in batch mode, they do not all process well by CM. One of the considerations is that a batch process requires homogeneity over space, whereas a CM process requires it over time [19]. Thus, the segregation of material in a feed hopper (e.g., due to a wide particle size distribution) can lead to process variation. Furthermore, while dead zones and sticky residues are primarily a cleaning and change-over issue in batch processing, they can lead to poor material tracking and inter-batch contamination in CM.

A number of suppliers have produced excipients tailored to CM, and in particular BASF, DFE Pharma, DOW, JRS Pharma and Roquette, have been mentioned in this regard [78].

The USP discussion guide on CM lists material properties, their characterising technique and the relevant USP Chapter on the use of the technique [99]. A paper by the FDA discusses the establishment of a material library for CM processes [105] with a particular focus on bulk flow properties. A total of 20 pharmaceutical materials were characterised by 44 properties capturing 880 data points. Measurements included PSD, flow properties with a Freeman Technologies FT4 rheometer, compressibility, permeability, shear cell, dynamic flow, etc. The data were processed by multivariate analysis to allow clustering. Feed performance (consistency and accuracy) was tested in a common feeder using three different types of screws and a volumetric (constant screw speed) setting. They found that a sub-set of measurements, namely, compressibility test, permeability test, shear cell test at 3 kPa and shear cell test at 9 kPa was able to identify the behaviour clusters and predict feeder performance. The authors stress that these measurements only characterise behaviour in this unit operation.

The importance of choosing the correct grade of excipient in a CM operation is highlighted in a recent paper by Allenspach et al. [107]. They found that different grades of HPMC showed different tribocharging properties when fed into a CM process from a loss in weight feeder. Thus the controlled release (CR) grade built up an electrostatic charge, which meant it stuck to parts of the feeder, leading to mass flow excursions and failures during refill. The direct compression (DC) grades did not show this. Interestingly the location of the electrostatic charge build-up point was different if the feeder was off-line (discharging into a bucket) or was connected to the CM system.

Direct compression tabletting is the simplest CM process, and excipients have been optimised to facilitate this. An API’s properties, also dependent on its dose, may mean it has poor flow and compressibility characteristics and hence make it unsuitable for DC. This forces manufacture into the more complicated dry or wet granulation processes. A recent paper from CMAC [108] has proposed that co-processed APIs could be produced that are suitable for DC CM. Specifically, diverse opportunities to significantly enhance API physical properties are created if allowances are made for generating co-processed APIs by introducing nonactive components (e.g., excipients, additives, carriers) during drug substance manufacturing. This would then facilitate E2E manufacturing. The paper gives details on the various types of co-processed API which might be considered and describes how their production is more easily achieved as part of the drug substance process as drug product facilities often do not have the necessary equipment (solvent handling, etc.). The regulatory position on these materials is presently unclear. Classifying them as drug substances has advantages with regard to re-work potential, primary stability and re-test dating expectations and timings for the start of drug product manufacture and hence shelf-life assignment. The paper details what testing and specifications might be required to have these classified as drug substances.

## 8. Industry/Academia/Government Collaborations to Progress Continuous Manufacturing

### 8.1. International Symposia for Continuous Manufacture of Pharmaceuticals (ISCMP)

These symposia are organised jointly by MIT (Cambridge, Mass, USA) and CMAC (Strathclyde, UK) and have been held in 2014, 2016, 2018 and virtually in 2021 [18,109,110,111]. They are the major CM symposia and provide a forum for discussions between industry, academia and regulators. Their output has helped shape progress in CM.

At the first symposium, Janet Woodcock emphasised the FDA’s support for CM and detailed its advantages. She described challenges to its implementation as being appropriate measurement and control systems, the ability to track material through the process, determining when to collect the product in specification (and rejecting product which is not) and acknowledged that control systems would need to be different. Several white papers were issued from the symposium (see J. Pharm Sci edition including Myerson et al. [97]), the most important of which detailed the regulatory and quality considerations for CM [45]. This paper details the guidelines current at the time, which, while not specifically citing CM, were supportive of the manufacturing method. It goes on to describe the quality requirements specific to CM.

A further paper [14] discussed the supply chain opportunities and challenges arising from the implementation of CM. It details the advantages of an end-to-end CM process, claiming that implementation would lead to a reduction in cycle time (starting materials to packed products) of 50% and reduced drug development costs of 10%.

While the first symposium was exclusively on small molecules, the second also included biopharmaceuticals. Significant progress was recognised between the two symposia (2014 to 2016) with ‘nearly all major innovator pharmaceutical companies working on continuous manufacturing technologies’ and two CM products approved by the FDA. The white paper issuing from this meeting [19] focussed on the regulatory aspects of CM, including the various initiatives that each regulatory authority had established to support and encourage the implementation of CM. It detailed the aspects of CM (process dynamics including residence time distribution, control strategy, process monitoring and control, material collection and diversion, real-time release testing and process validation and verification) which need to be considered. The ideas in this and the previous white paper will be key in helping shape the proposed ICH Q13 guideline.

There was no white paper issued from the third symposium in 2018, but details of further progress were presented [100]. The FDA, again Janet Woodcock, accepted that there is not a business case for CM for every product and cautioned that CM must not be used as a way of shutting out generics or biosimilar competition. She emphasised the importance of collaboration to advance the field and specifically spoke of the NIIMBL consortium (see Section 8.2.8). This point was also taken up by a representative from Janssen (one of the pharma companies that have commercialised a CM product) who suggested that pharma companies could help each other by sharing their understanding of control strategies, equipment, risks and impacts of material variability, process and environment.

A useful presentation at the 2018 meeting reviewed the progress and shared the experiences of 13 pharma companies in CM [112].

The fourth Symposium was held virtually on 18 February 2021 and focussed primarily on the UK implementation of CM [18].

### 8.2. Consortia Working to Progress CM

The industry has taken a cautious approach to the implementation of CM. Working in consortia and public-private partnerships can reduce the risks associated with CM implementation [86]. Implementation of CM by generic companies can be particularly difficult. A model using manufacturing hubs based at universities or in CDMOs could provide expertise in CM and handle projects for both generics and innovators [8]. The following sections detail some of the existing collaborations.

#### 8.2.1. Novartis—MIT Center for Continuous Manufacturing 

Founded in 2007, this centre [113] based in Cambridge, MA, USA, was established as a 10-year partnership and had the objective of transforming pharmaceutical production. It developed an end-to-end CM unit capable of turning raw ingredients into finished tablets, and details were published [93]. In 2013, a spin-out, Continuus Pharmaceuticals [114], was launched, and in 2021 they received a US Government contract to develop a GMP end-to-end plant to manufacture three medicines by CM [94].

#### 8.2.2. CMAC

CMAC [115], based in Strathclyde, UK, is a world-class international hub for manufacturing research and training. Working in partnership with industry, its purpose is to transform current manufacturing processes into the medicine supply chain of the future. It consists of seven UK universities and eight major pharma companies working together with other partner companies and organisations. Most output from CMAC has been on CM relating to API synthesis, and in particular, API crystallisation; however, their interests extend to end-to-end supply chains, improving understanding of material attributes, computer-aided process design and the concept of micro-factories.

CMAC is involved with the CPI—Medicines Manufacturing Innovation Centre and is a founder member of the International Institute for Advanced Pharmaceutical Manufacturing (IIAPM) (see Section 8.2.5).

#### 8.2.3. C-SOPS

Founded in 2006, the Center for Structured Organic Particulate Systems (C-SOPS) [116] is headquartered at Rutgers University and partners with Purdue University, New Jersey Institute of Technology and the University of Puerto Rico at Mayaguez. Its aim is to develop science and engineering methods for designing, scaling, optimising and controlling dosage forms and relevant manufacturing processes. It is a founder member of the IIAPM (see Section 8.2.5).

Technology developed at C-SOPS was used by Janssen (part of J&J) for the commercialisation by CM of Prezista^®^, approved by the FDA in 2016.

#### 8.2.4. RCPE

The Research Centre for Pharmaceutical Engineering (RCPE) [117] is based in Graz, Austria, and is a global leader in pharmaceutical engineering sciences, encompassing continuous API synthesis, advanced formulations, next-generation manufacturing (including CM) and device design and optimisation. It partners with several pharma companies and others in the Center for Continuous Flow Synthesis and Processing (CC FLOW) for the continuous manufacture of APIs. It is a founder member of IIAPM (see Section 8.2.5).

#### 8.2.5. IIAPM

The International Institute for Advanced Pharmaceutical Manufacturing (IIAPM or I2APM) [118] is a collaboration between CMAC, C-SOPS and RCPE and has the objective of advancing the science and technology of integrated primary and secondary continuous manufacturing of pharmaceutical products by promoting pre-competitive research. It organises meetings, cooperates internationally on projects and provides training for industry, researchers and regulators.

#### 8.2.6. MMIP/MMIC

MMIP (Medicines Manufacturing Industrial Partnership) is a UK-based group working between government and industry to champion and facilitate manufacturing for medicines in the UK. Continuous manufacturing is one of the components of its technology roadmap.

Lobbying from the MMIP (and others) led to funding for the creation of the UK Centre for Process Innovation Medicines Manufacturing Innovation Centre (CPI-MMIC) is a GMP centre being built near Glasgow, UK, due for completion late 2021 [119]. It is a collaboration between CPI, the University of Strathclyde, UK Research and Innovation, Scottish Enterprise and founding industry partners GSK and AstraZeneca. Other organisations are also involved. Its aim is to provide a facility that will enable industry, academia, healthcare providers and regulators to work together collaboratively to address challenges and maximise technology opportunities within the medicines supply chain.

One of its grand challenges (working in partnership with CMAC) is CM of direct compression (DC) solid oral dosage forms. Plans are:To create a ‘digital twin’ of the CM process and via virtual formulation experiments to select equipment and initial processing conditions for CM and thereby give confidence that CM DC process will work. This will be useful evidence that a company might use to justify an investment in CM.To enhance understanding of raw materials to build a CM control strategy to allow RTRT and develop robust formulation processes on a modular flexible CM DC platform.To de-risk and accelerate CM DC technology by modularising equipment and standardising control systems moving away from supplier-specific approaches [120].

#### 8.2.7. SSPC

The Synthesis and Solid-State Pharmaceutical Centre (SSPC) [121] is a hub of Irish research expertise developing innovative technologies to address key challenges facing the pharmaceutical and biopharmaceutical industry.

In August 2020, the SSPC was awarded a 1.9M EUR grant to build a pharmaceutical manufacturing facility where academia will work with industry to develop CM processes [122].

#### 8.2.8. NIIMBL

The National Institute for Innovation in Manufacturing Biopharmaceuticals (NIIMBL) [123] is a US collaboration between industry and academia that has the mission to accelerate biopharmaceutical innovation, support the development of standards that enable more efficient and rapid manufacturing capabilities and educate and train a world-leading biopharmaceutical manufacturing workforce. In July 2019, they announced a Collaborative Research and Development Agreement (CRADA) with the Food and Drug Administration [124] to support investments in regulatory science research and training needed to foster advanced manufacturing innovations in areas such as continuous manufacturing, on-demand manufacturing and advanced process control technologies.

#### 8.2.9. UK-CPI National Formulation Centre, National Industrial Biotechnology Centre and National Biologics Manufacturing Centre

These organisations provide equipment and technical expertise that help companies develop next-generation products, including the use of CM [125].

#### 8.2.10. AMED

AMED (Japan Agency for Medical Research and Development) [126], established in 2015, is a Japanese collaboration between the Japanese Regulator PDMA and industry aimed at promoting integrated research and development in the field of medicine, from basic research to clinical trials. Their project for Advanced Drug Discovery and Development includes manufacturing technology, and they have published a number of useful papers on CM (see elsewhere in this paper).

### 8.3. Other Supportive Organisations and Guidance

#### 8.3.1. United States Pharmacopeia

The USP [127] has recently published as a stimulus to the revision process a document on CM [99]. This document defines the various terms associated with CM, the requirements to characterise materials used in CM processes with details of the various material attributes, which might be studied and a reference to relevant USP-NF chapters where these exist. Tables detail the various potential failure modes of a simple CM process (unit operations consisting of feeders, mill, blender and tablet press) and which material properties might be critical in these. A further chapter covers risk management, PAT and statistical tools. A section on regulatory considerations and industrial adoption of CM acknowledges that take-up has been slow and provides some of the reasons together with successes and products using CM commercialised to date. A table sets out the key elements of a CM process and regulatory expectations for each of these.

The USP is also working with C-SOPS to develop training courses on standards associated with CM [8]. In February 2021, they announced a strategic alliance with the US Phlow Corporation to develop a new laboratory that will certify and validate pharmaceutical CM processes. The aim is to encourage domestic generics to adopt CM and thereby strengthen the US drug supply [128].

#### 8.3.2. ASTM

The American Society for Testing and Materials (ASTM) [129] has published guides on continuous processing in the pharmaceutical industry, PAT-enabled control systems and continuous process verification [97,130,131].

#### 8.3.3. ISPE

The International Society for Pharmaceutical Engineering (ISPE) [132] has organised conferences on CM [133] and has published a number of useful articles, including on sampling considerations in continuous manufacturing and on process validation of small molecule drug substance and drug product CM [134,135].

#### 8.3.4. NIPTE

The National Institute for Pharmaceutical Technology and Education (NIPTE) [136], based in the US, has the mission to improve the way medicines are designed, developed and manufactured to meet the needs of patients in the twenty-first century. The NIPTE has advocated the sharing of critical information on quality issues and potential failure modes for CM.

## Figures and Tables

**Figure 1 pharmaceutics-13-01311-f001:**
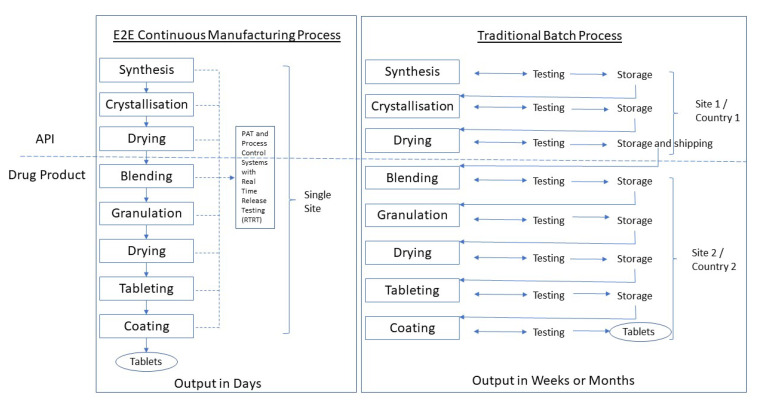
Schematic of a continuous manufacturing end-to-end (E2E) process compared to a traditional batch process. Adapted from [3], J. Pharm. Innov. 2015. Note: API manufacturing has traditionally occurred at a primary manufacturing site typically separate from the drug product site and often in a different (low-tax) country.

**Figure 2 pharmaceutics-13-01311-f002:**
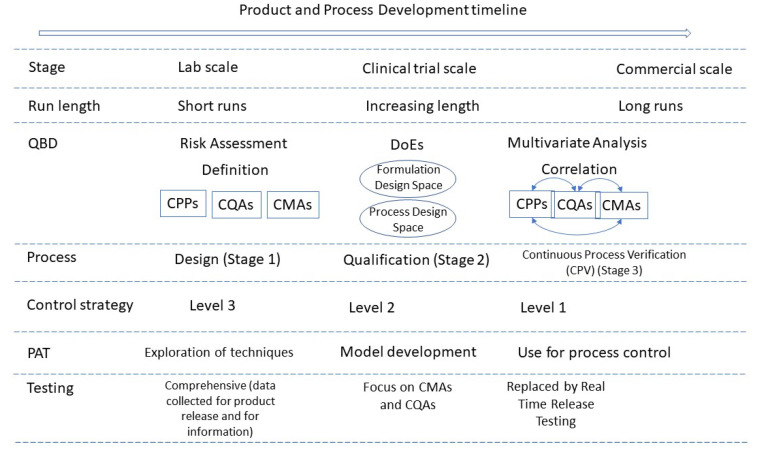
Schematic of continuous manufacturing process development. Adapted from [68], Pharmaceutics 2021. DoEs = Design of Experiments. Schematic assumes scale-up is achieved by running CM process for longer periods and technology transfer is avoided by using same equipment throughout development and into commercialisation.

**Table 1 pharmaceutics-13-01311-t001:** Approved drug products manufactured by continuous manufacturing (CM) Adapted from [4,5].

Drug Product	Indication	Company	Year of First Approval	Regulatory Body
Orkambi^®^	Cystic fibrosis	Vertex	2015	EMA, FDA
Prezista^®^ *	HIV	Janssen (J&J)	2016	EMA, FDA
Verzenio^®^	Breast cancer	Eli Lilly	2017	EMA, FDA, PDMA
Lorbrena^®^	Lung cancer	Pfizer	2018	**
Daurismo^®^	Myeloid leukaemia	Pfizer	2018	FDA
Symkevi^®^/Symdeko^®^	Cystic fibrosis	Vertex	2018	EMA/FDA
Tramacet^®^	Pain	J&J	**	PMDA

* Batch to CM conversion others are NCEs. ** Information not available.

**Table 2 pharmaceutics-13-01311-t002:** Benefits of adopting CM.

Faster and leaner transition from development to commercial scale	Same equipment used during development, clinical supply manufacture and commercial production with batch sizes accommodated by changing run times Shorter cycle times More rapid development
Shorter supply chains	All operations conducted on one piece of equipment in one location without hold-ups and inter-site transfers between manufacturing stages Improved stability as no intermediate holding periods
Supply chain security	Enhanced domestic manufacture with reliance on high technology rather than low-cost labour Shorter manufacturing times meaning longer product shelf lives
Improved product quality	Domestic manufacture In-process monitoring, feedback and feed-forward controls. Maintenance of a state-of-control Reduced dependence on end-product testing Decreased regulatory oversight—frees resources for other higher-risk areas Fully aligned with principles of Quality by Design (QbD)
Cost benefits (after initial investment)	Lower production costs Improved utilisation of equipment Lower personnel requirements Smaller footprint
Supply chain responsiveness	Batch sizes tailored to requirements by adjusting run times Ability to respond rapidly to demands
Patient benefits	More suited to niche/personalised products Alternative manufacturing technologies (e.g., active pharmaceutical ingredient (API) printed or sprayed onto a dosage form) Lower risk of stock-outs
Societal benefits	Less environmental impact (less use of solvents, lower energy costs) Less waste/improved yields. Lowered risk that a whole batch will need to be rejected if it fails end-product testing Higher process intensification (less use of space, energy and raw materials) Improved safety (reduced handling and exposure to materials, easier cleaning) Source of high-tech jobs

Extracted and adapted from [3,4,9,10,11].

**Table 3 pharmaceutics-13-01311-t003:** Challenges to implementing CM.

Existing equipment and facilities are geared to batch processes	Initial investment cost to implement CM (albeit with lower subsequent costs) Existing equipment likely depreciated so may be less of an issue
Facilities not located to achieve end-to-end processing	API and drug product facilities may be in different countries (perhaps driven by tax benefits) or locations within a country
Different expertise requirements	CM requires experts in statistics, process control, modelling, QbD processes, PAT (Process Analytical Technology), etc. Requirement to better understand material attributes
CM process	Limited possibilities for re-work (but also less likely a need to do so) Limited opportunities to halt the process part way through
Maintenance	Control algorithms and models require adjustment to accommodate changes in raw materials. May have regulatory implications Sophisticated PAT equipment
Submission requirements	New submissions required to switch batch to CM No globally recognised harmonised CM approval process Concern that regulators will baulk at CM processes Limited expertise in non-FDA, EMA, MHRA, PMDA regulated countries Longer approval times for global registration
Equipment	Lack of appropriate bench and pilot-scale equipment Not all unit operations can currently be included in a CM process Handling of dry solids and solid laden fluids can be difficult
Experience	Lack of examples of end-to-end processes Limited company experience of CM submissions Limited experience in regulators (in particular EMA, MHRA and PMDA)

Extracted and adapted from [5,9,11,13,14].

**Table 4 pharmaceutics-13-01311-t004:** Issued ICH guidelines of relevance to CM for a drug product, adapted from [20,21,22,23].

Number	Date	Title	Contents of Relevance to CM
ICH Q8(R2)	2009	Pharmaceutical Development	Control strategy, continuous process verification
ICH Q9	2015	Quality Risk Management	Risk assessment and control
ICH Q10	2008	Pharmaceutical Quality System	Continual improvement of process performance and product quality
Quality-IWG	2012	Points to consider for ICH Q8/Q9/Q10 guidelines	Models in QbD, continuous process verification

## Data Availability

Not applicable.

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
