# Peer review of "(untitled)"

_pharmaceutics, 2021, doi:10.3390/pharmaceutics13081311_

Round 1
Reviewer 1 Report
This manuscript reviews continuous manufacture of solid oral dosage forms. The aim and scope of this manuscript would be interesting to the readers of the journal. However, the contents of this work should be improved using more detailed, relevant and practical examples of previous research and/or industrial processes. Also, the manuscript needs to be extensively re-organized using high-quality schemes, figures, and tables to facilitate the readers' understanding on this subject. Thus, the manuscript in the present form is not suitable for publication in a good-impact journal like Pharmaceutics.
Author Response
The aim of this review is to provide a general introduction to continuous manufacturing and to highlight some of the more recent developments. Addition of practical examples would likely add a level of detail beyond a general introduction. The article is well referenced allowing the reader to find detailed examples. Two figures have been added to clarify aspects of the review as suggested by the reviewer. The review already contains 4 tables. With regards to the comment that the review is not suitable for the journal I refer to the other reviewers who suggest it is appropriate for publication.
Reviewer 2 Report
The authors wrote a very extensive and detailed review on the continuous manufacturing of small molecules for drugs. They also take into very consideration the current regulations required for these kind of products. I found it very helpful as a guide and general concept. My only concern is that there are no citations of this year 2021. Therefore, my recommendation is to publish it with more references of this year.
Author Response
There were already 3 2021 references. An additional 6 2021 references have been added in the revised paper.
Reviewer 3 Report
In the current article the authors present a review on Continuous Manufacturing of Small Molecule Solid Oral Dosage Forms. The topic is timely and the article is well written with up to date literature. The article provides a comprehensive understanding to the readers and those familiar with Continuous Manufacturing processes.
I recommend the acceptance of the manuscript
Author Response
No comments
Reviewer 4 Report
Continuous manufacturing technology has become more and more prominent in various industries. There is a growing interest toward the production lines with innovative solutions suitable for continuous production in the pharmaceutical industry, too. Some commercially available formulations are now manufactured using continuous manufacturing technology. However, it is a real challenge for the developer/ manufacturer to make the traditional batch mode continuous.
Thus, it can be concluded that the manuscript can be considered as a comprehensive review, which is topical/actual and valuable for many readers. In particular, the manuscript deals with the regulation of continuous production and presents several factory solutions (ConsigmaTM; Granucon®, QbCon®; Granuformer®).
Some critical remarks for the improvement:
- No figure was found in the manuscript. It can even be a schematic representation of a batch and a continuous manufacturing process for tablets.
- The abbreviation E2E also appears on lines 32 and 49 but is only resolved on line 568.
- Some references from the internet need retrieval date.
Author Response
Two figures have been added to the revised manuscript.
E2E was explained at first mention in the original manuscript. One of the new figures adds a further explanation.
All of the internet references have access dates.
The reviewers scoring system doesn't seem to match their comments and recommendation to publish.